# Evaluation of Inappropriate Prescribing in Patients Older than 65 Years in Primary Health Care

**DOI:** 10.3390/jcm8030305

**Published:** 2019-03-04

**Authors:** Antonio Nuñez-Montenegro, Alonso Montiel-Luque, Esther Martin-Aurioles, Felicisima Garcia-Dillana, Monica Krag-Jiménez, Jose A. González-Correa

**Affiliations:** 1UGC Primary Health Care, Biomedical Research Institute of Malaga-IBIMA, Primary Health Care Centre Archidona, Health Area North of Málaga, 29011 Málaga, Spain; 2Primary Health Care Centre San Miguel (Torremolinos), Health District Costa del Sol, 29620 Málaga, Spain; alonsomontiel@uma.es; 3UGC Primary Health Care, Biomedical Research Institute of Malaga-IBIMA, Primary Health Care Centre La Roca, Health District Málaga-Guadalhorce, 29011 Málaga, Spain; estherd.martin.sspa@juntadeandalucia.es; 4Primary Health Care Centre Cala de Mijas (Mijas), Health District Costa del Sol, 29649 Málaga, Spain; feli.garcia.dillana@gmail.com; 5Primary Health Care Centre Fuengirola Oeste (Fuengirola), Health District Costa del Sol, 29640 Málaga, Spain; monica.krag.sspa@juntadeandalucia.es; 6Department of Pharmacology, Biomedical Research Institute of Malaga-IBIMA, School of Medicine, IBIMA-University of Málaga, 29071 Málaga, Spain; correa@uma.es

**Keywords:** elderly, inappropriate prescribing, patient safety, polymedicated, primary care, risk factors STOPP/START

## Abstract

To asses inappropriate prescribing and its predisposing factors in polymedicated patients over the age of 65 in primary health care. Design: cross-sectional study. Setting: Primary care centres in the Costa del Sol Health District and Northern Health Area of Malaga in southern Spain. Participants: Patients older than 65 years who use multiple medications. Data collection was conducted during 1 year in a population of 425 individuals who comprised a stratified randomized sample of the population of health care users in the study area. The data were collected by interview on a structured data collection form. Study variables. Dependent variable: Potentially inappropriate prescribing (PIP) (STOPP/START criteria). Predictor variables: Sociodemographic characteristics, clinical characteristics and medication use. A descriptive analysis of the variables was performed. Statistical inference was based on bivariate analysis (Student’s *t* or Mann-Whitney *U* test and chi-squared test) and multivariate analysis was used to control for confounding factors. 73.6% of participants met one or more STOPP/START criteria. According to information about prescribed treatments, 48.5% of participants met at least one STOPP criterion and 43.30% of them met at least one START criterion. The largest percentage of inappropriate prescriptions was associated with cardiovascular treatments. More than three-quarters of the participants had one or more inappropriate prescriptions for medicines in primary care, according to STOPP/START criteria. In addition, PIP was directly related to the number of prescribed medications, gender and specific pathologies (diabetes).

## 1. Introduction

Among older patients, 80% suffer from chronic illness. This group of users generates a three-fold higher demand for care than the general population, consumes a significant percentage of medicines and accounts for 75% of pharmaceutical expenditures [1,2,3]. In Spain, 13.86% of the population over the age of 65 years is polymedicated, that is, they use 5 or more medicines for 6 months or longer. Prescribing medicines for these patients is complex and some problems could be avoided or minimized by a more efficient use of medicines. Appropriate prescribing is characterized by clear evidence supporting the use of a medicine for a specific indication, good tolerance and a favourable cost-effectiveness profile [3,4]. In contrast, potentially inappropriate prescribing (PIP) occurs when the clinical benefits do not outweigh the risk of side effects. In other words, unsuitable medicines may be prescribed, for more frequent or longer dosages than indicated in the patient information leaflet. Other types of PIP involve a high risk of medicine-medicine or medicine-disease interaction or duplicate medications with the same mechanism of action. Furthermore, PIP also includes the non-prescription of indicated drugs. Inappropriate prescribing of medicines is a problem that affects older patients who are polymedicated and have multiple comorbidities and is also a major cause of morbidity and mortality [5,6,7,8].

Studies published to date have established the prevalence of PIP in the population of older polymedicated people. However, patient-related factors can influence the therapeutic approach chosen by the patient’s physician. Therefore, the aim of this study was to determine the frequency of inappropriate prescribing by primary care physicians and its predisposing factors in patients who used multiple prescription drugs and were older than 65 years.

## 2. Methods

The study used a cross-sectional descriptive design.

The study participants were polymedicated patients over the age of 65, treated at different primary care centres located in the Costa del Sol Health District and the Northern Health Area of Malaga (19 clinical management units in all) in Spain. The patients were selected from lists of polymedicated patients older than 65 years by stratified randomized sampling. The lists of polymedicated patients at each of the 19 clinical management units were provided by the pharmacy service of each health district.

The sample size was calculated by using the percentage of polymedicated patients older than 65 years with PIP (estimated prevalence of 50%). A sample of 425 individuals was necessary to achieve 5% estimation accuracy for bilateral analysis and a 95% asymptotic confidence interval for a population of 12,366 individuals, assuming a 10% loss to follow-up.

Inclusion criteria

Polymedication (use of 5 or more medications for a period of ≥6 months) and age 65 years or older.

Inclusion in the electronic prescription program.

Exclusion criteria

Patients with functional or cognitive impairment that impeded autonomous management of their medication (Barthel Index ≤60 or more than 4 errors in the Portable Mental Status Questionnaire (SPMSQ) (Pfeiffer test).

Institutionalized patients or those with a psychiatric illness.

Refusal to provide written informed consent.

Language barriers.

Data collection

One study leader was appointed to collect data in each health district, area and centre. Once stratified randomized sampling was completed, the study leaders at each centre or district began recruiting participants.

Of note, the individual responsible for data collection was the same for both districts. The interviews took place at the patient’s home. The data collection form designed for the purpose of this study contained all the tests and variables to be measured during the interview. Face-to-face interviews were done at the patient’s home using a specifically-designed questionnaire that included all tests and variables to be assessed in the study. On-site evaluations of medicine cabinets were also done, in order to check the medication that really consumed the patient. In addition, the prescriptions included in the electronic prescription program were reviewed by two health professionals.

Study variables

Dependent variable: Inappropriate prescribing (STOPP/START criteria). The STOPP-START criteria (Screening Tool of Older Person’s Potentially Inappropriate Prescriptions/Screening Tool to Alert Doctors to the Right Treatment) collect the most common errors of treatment and omission in prescription in the elderly. The STOPP-START criteria have proven to be a good tool for detecting potentially inappropriate prescriptions and improving the quality of prescription in elderly people in all care settings. (added in the text).

A single researcher was responsible for reviewing and coding the stop/start criteria. These criteria, organized by physiological systems, can be applied quickly (in approximately 10–15 min), collect the most common errors of treatment and omission in the prescription and are easy to relate to the active diagnoses and the list of drugs that appear in the histories patient’s computer clinics.

Independent variables

Sociodemographic: Age, gender, residence, cohabitation, education, social risk assessment (Gijón Scale).

Clinical data: Diseases and disorders (Pathologies and processes collected in the computerized medical record for which the patient received the medication), functional assessment (Barthel Index), cognitive assessment (Pfeiffer test), screening assessment of anxiety and depression (Goldberg Anxiety and Depression Scale (GADS) and number of emergency room visits (Number of times the patient went to the emergency department, in relation to decompensation of his disease or appearance of acute pathology).

Medication: Number of medicines (Number of prescribed medications that the patient consumes daily), number of prescribing doctors (Number of doctors, primary care physician or specialist physician, who have prescribed a medication to the patient), adherence to treatment (assessed with the Morisky-Green-Levine Medication Adherence Scale (MGLS).

Quality of life: The validated Spanish translation of the EuroQol-5 Dimension instrument (EQ-5D), a generic health-related quality of life questionnaire, was used. This instrument is widely used in primary care.

All those data were easily extractable from the history of digital health. The diagnoses are codified, the emergency assistance can be consulted in the computerized history, in the medication sheet are the prescribed drugs and the doctor who performs the prescription.

### 2.1. Statistical Analysis

The data were analysed with SPSS software v. 23.0 (SPSS, Chicago, IL, USA). Quantitative variables are expressed here as the mean and standard deviation and qualitative variables are reported as frequencies and percentages. Bivariate analysis was done with Student’s t-test (or the Mann-Whitney *U* test) and chi-squared tests were used for qualitative variables. Multivariate analysis was done with binary logistic regression. The predictive variables included in the binary logistic regression model, in addition to age and gender, were those that were associated with inadequate prescription in the bivariate analysis. The variables were: cohabitation, social assessment, mental assessment, Barthel Index, GADS, main pathologies, emergency room, number of medicines, number of prescribers. Forward method was used for the analysis.

### 2.2. Ethical Considerations

This study was approved by the Ethics and Research Committees of the Northern Health Area of Malaga and the Costa del Sol Health District, which verified that the study was performed in accordance with all ethical standards set forth by the committees, as well as by the Declaration of Helsinki (Fortaleza 2013).

## 3. Results

Among the key demographic characteristics of the sample, the average age of the respondents was 74.68 ± 5.62 years and 64.2% of them were women. Moreover, 20.2% of the patients lived alone. According to the Gijón Scale, only 15.8% of participants had an intermediate social risk (Table 1).

The most frequent chronic diseases in our population were hypertension (84.9%), dyslipidaemia (63.1%), heart disease (53.4%) and diabetes (52.7%). Of note was that patients in our sample had a 60% risk of anxiety and a 32.2% risk of depression, respectively. Regarding the use of health resources, 49.2% of our participants made between 1 and 3 emergency room visits in the previous year and 6.1% made more than 3 visits to emergency services (Table 1).

Nearly half (48.5%) of the participants used more than 10 medicines. Inappropriate prescribing was detected in 45.2% of the participants and no significant differences were observed in terms of age (*p* = 0.271) or sex (*p* = 0.892). Medication errors of any type (omission, duplication, dose, frequency) were detected in 82.6% of the cases analysed here (Table 1).

Nearly three fourths (73.6%) of the participants met at least one STOPP/START criterion. In the breakdown of this information, 48.5% of the participants met one or more STOPP criteria for prescribed treatments. The most frequent medications were prescribed for the cardiovascular (16.9%), gastrointestinal (15.5%) and musculoskeletal system (15.3%) and for medication-related central nervous system treatments (10.8%).

One or more START criteria were identified in 43.30% of the participants. The most frequent criteria were related to the cardiovascular systems (16%) and 1.2% were related with the endocrine system (22.4% according to STOPP/START version 1 criteria). The frequency of STOPP criteria decreased slightly with age, while the opposite trend was seen for the START criteria. Concerning gender, women had a higher percentage of inappropriate prescribing: 78% versus 65.8% (*p* = 0.006) (Table 2).

With regard to pathologies, inappropriate prescribing was most frequent among patients with a diagnosis of diabetes (Table 2). We found a statistically significant relationship between the number of prescribed medicines and the presence of STOPP criteria (*p* = 0.017). A similar relationship was seen for the START criteria, although this relationship was not statistically significant. We found no relationship between the presence of STOPP/START criteria and adherence. On the other hand, we observed a higher percentage of PIP in patients with a higher frequency of emergency room attendance (Table 2).

Worse quality of life was related to inappropriate prescribing (58.08 vs. 64.14; *p* = 0.009) and this relationship was more evident for STOPP criteria. The factors related with inappropriate prescribing are reported above and those specially affecting any of the STOPP or START criteria are also noted. No statistically significant differences between genders were seen in the frequencies of either set of criteria. However, we noted a greater number of START criteria in patients with diabetes and at risk of depression, while higher numbers of medicines were associated with STOPP criteria (Table 2).

Multivariate analysis showed that the variables female gender, diagnosis of diabetes and use of more than 10 medicines were directly related to inappropriate prescribing (Table 3).

## 4. Discussion

Our results, like those of earlier studies, document the high prevalence of PIP (73.6%) in polymedicated patients older than 65 years. However, the results across studies vary depending on the methodology and criteria used, with frequencies ranging from 24% to 73% [6,8,9].

We chose the STOPP/START criteria not only for their added value in detecting inappropriate prescribing of certain medicines but also because they can flag the absences of necessary and important drug prescriptions for the treatment of some processes [4]. We used these criteria together with review of our participants’ medical records and the contents of their medicine cabinets, an approach which avoided overestimation of PIP—a potential drawback of studies based on review of drug lists [10]. We found that 48.5% of participants met some STOPP criteria and 43.30% met some START criteria; these percentages are similar to those reported in other studies [11]. The STOPP-START criteria have proven to be a good tool for detecting potentially inappropriate prescriptions and improving the quality of prescription in elderly people in all care settings. In order to set well-defined criteria and fill in the gap in this field, geriatric pharmacotherapy specialists from Ireland and the United Kingdom accepted START/STOPP criteria on the basis of the Delphi consensus method in 2008. STOPP involves a 65-item list with drug-drug and drug-disease interaction, therapeutic duplication and increased risk of cognitive deterioration. On the other hand, START is a list of 22 rules related to common instances of prescribing omissions for older people. These criteria apply to patients aged over 65.

We observed the highest percentage of inappropriate prescriptions for medications to treat cardiovascular diseases. In our patient sample, 16.9% met STOPP criteria and 16% met START criteria for this group of drugs—figures similar to those in other studies [1,11,12,13,14]. The second most frequent group of inappropriate prescriptions was for medications to treat musculoskeletal illnesses, a result strongly related to the use of anti-inflammatory medicines [7]. According to our analysis of the main criteria, we found that antiplatelet drugs were the principal cause of PIP, either because of inappropriate doses (6.7%) or as a default prescription (5.6%). These percentages, again, are similar to other studies [7,15,16]. Another important criterion related to the frequency of occurrence was the non-use of statins (7.5%) in patients with cardiovascular disease. Compared to other studies such as that by Filomena et al. [16], we noted that our PIP was much higher for statins.

There is a known relationship between the use of benzodiazepines and the risk of falling. In patients with osteoporosis in the present analysis, we found a 9.2% rate of PIP for benzodiazepine use. According to the STOPP/START criteria, we noted a 7.8% rate of omission of prescriptions for calcium and vitamin D supplements [7,15,17].

The predominant social profile of our patients with inappropriate prescriptions was women over 70 years old, living with others (usually with a partner), with intermediate socioeconomic status, low educational level and low social risk. Other research has reported a higher risk of PIP (OR (Odds Ratio) = 1.16) in women regardless of other social factors [18].

Regarding the number of medicines, most participants in our sample used more than 8 medicines. In our study, as in others, the use of potentially inappropriate prescriptions is related directly to the number of medicines used, hence a greater number of medicines is associated with an increased frequency of STOPP criteria [1,7,19,20]. In addition, we observed a connection between inappropriate prescribing and the use of health resources. In this regard, our findings confirmed that users who sought health services more often were also more likely to meet the criteria for inappropriate prescriptions [21].

The average score of perceived quality of life in our participants, as measured by the EQ-5D instrument, was within the range of values reported by other studies in Spain [22,23]. Of note, however, is our finding of an association between poorer quality of life and some STOPP or START criteria. The few relevant articles published thus far have not noted an influence of PIP on quality of life [24].

We have used the most commonly used tools in our environment. We did not propose a clinical diagnosis of depression that would already exist in the patient’s history but the existence of symptoms suggestive of anxiety / depression.

Goldberg scale was used for participants in this study. It should be noted that this scale is not a diagnostic instrument but rather a scale used to detect the risk of depression and/or anxiety. The clinical significance of the Goldberg scale lies in its ability to identify patients at risk for depression and/or anxiety symptoms—an important consideration for clinicians during follow-up of these patients.

On the other hand, we use the EuroQol-5D instrument, a generic HRQoL questionnaire for the evaluation of quality of life in polymedicated patients. The model used consisted of three parts, one of them (EQ-Index) describes the health status according to five dimensions (mobility, self-care, usual activities, pain/discomfort and anxiety/depression). In our study anxiety-depression were the most prevalent disorders (69.1%).

Polypharmacy is common in older patients with diabetes mellitus, in whom PIP rates are high even when medications used to treat diabetes are excluded [25]. The profile of a person with inappropriate prescribing is a patient with diabetes, use of more than 10 medicines and female gender. Knowledge of the profile of patients with inappropriate prescribing will make it possible to design and implement appropriate strategies for them.

One of the main limitations of this study is its cross-sectional design, which precludes the establishment of direct causal relationships. However, the identification of factors related to PIP, which was one of our intended goals, was well within the capabilities of this type of study.

We think that the inclusion criteria do not have biases. The exclusion of patients with dementia or psychiatric illness, as well as language barriers, were made worse by the interview with patients. Although the exclusion of patients with dementia or psychiatric illness can be considered a bias, (higher prevalence of "PIP" in these patients), they represent a small percentage of the total number of polymedicated patients.

Due to the design of our study, we can only affirm that being a woman, the number of prescribed medications and diabetes, is related to a higher prevalence of PIP. We cannot assume the predictive nature of these variables on the detection of inadequate prescription.

## 5. Conclusions

In conclusion, more than three-quarters of the participants in our analysis received inappropriate prescriptions for drugs according to STOPP/START criteria. In addition, PIP was directly related to the number of prescribed medicines, gender and certain pathologies (diabetes). The greatest risks of inappropriate prescribing were associated with medications widely used in clinical practice for cardiovascular, musculoskeletal, endocrine, gastrointestinal and central nervous systems diseases and disorders. Bearing in mind that the STOPP/START criteria cannot substitute for clinical judgment, we present data that reveal the need for a review of treatment plans for polymedicated patients over the age of 65 years, as a simple tool against the excess and inappropriate prescribing detected in most studies. Further research is needed to evaluate health outcomes in patients with PIP and analyse the strategies needed to minimize this practice.

## Figures and Tables

**Table 1 jcm-08-00305-t001:** Study variables and description of the sample.

Variables (N = 425)	Categories	*n*	%
**Sociodemographic variables**			
Age (years)	65–69	98	23.1
	70–74	111	26.1
	75–79	124	29.2
	≥80	92	21.6
Gender	Male	152	35.8
	Female	273	64.2
Residence	Costa del Sol	327	76.9
	North Málaga	98	23.1
Cohabitation	Alone	86	20.2
	Accompanied	339	79.8
Level of education	Illiterate	83	19.5
	Literate	248	58.4
	Primary studies	20	4.7
	High school	43	10.1
	University studies	18	4.2
Social assessment *	Low o normal	358	84.2
(Gijón Scale)	Middle	67	15.8
**Clinical variables**			
Mental assessment	Normal	416	97.9
(Pfeiffer Test)	Mild impairment	9	2.1
BADL ^†^	Independent	298	70.12
(Barthel Index)	Mild dependence	82	19.29
	Moderate dependence	45	10.59
Emotional state	Anxiety	255	60
(GADS) ^‡^	Depression	137	32.2
Main pathologies	Hypertension	361	84.9
	Dyslipidaemia	268	63.1
	Heart disease	227	53.4
	Diabetes	224	52.7
	OCFA	64	15.1
	Stroke	32	7.5
ER ** visits	No	190	44.7
(last year)	1–3	209	49.2
	>3	26	6.1
**Medication variables**			
Number of medicines	5–7	64	15.1
	8–10	155	36.5
	>10	206	48.5
Number of prescribers	1 ó 2 >2	260 165	61.2 38.8
Medication adherence	Adherence	233	54.8
(MGLS) ***	Nonadherence	192	45.2
Medication errors	Yes No	333 92	73.4 21.6

* Social-familiar scale of Gijon (social risk): <10: normal or low; 10–16: intermediate; >16: high. **^†^** (Basic Activities of Daily Living). ** ER: Emergency Rooms. **^‡^** Goldberg Anxiety and Depression Scale: screening test for depression and anxiety; anxiety subscale (>4 risk of anxiety); depression subscale (>6 risk of depression). *** Morisky-Green-Levine Medication Adherence Scale.

**Table 2 jcm-08-00305-t002:** Prevalence of STOPP-START criteria.

Variables (N = 425)	Categories	Inappropriate Prescribing Frequency (%)	STOPP Frequency (%)	START Frequency (%)
**Gender**	Male Female	100 (65.8) * 213 (78)	65 (42.8) 141 (51.6)	60 (39.5) 124 (45.4)
**Diabetes**	No Yes	136 (67.7) 177 (79) **	98 (48.8) 108 (41.2)	62 (30.8) 122 (54.5) ***
**Number of medicines**	5–7 8–10 >10	44 (68.8) 102 (65.8) 167 (81.1) ***	25 (39.1) 65 (41.9) 116 (56.3) ****	25 (39.1) 61 (39.1) 98 (47.6)
**Risk of depression** **(GADS) ^1^**	No Yes	47 (16.3) 30 (21.9)	143 (49.7) 63 (46.0)	111 (38.5) 73 (53.3) *****

* *p* = 0,006; ** *p* = 0,008; *** *p* = 0,039; **** *p* = 0.0001, ***** *p* = 0.017; ^1^ Goldberg Anxiety and Depression Scale.

**Table 3 jcm-08-00305-t003:** Variables associated with Inappropriate Prescribing (logistic regression).

Variables	Inappropriate Prescribing		
	Odds Ratio	IC (95%)	*p*
Diabetes ^†^	1.847	1.173/2.910	0.008
Number of medicines ^‡^	1.990	1.260/3.142	0.003
Sex ^§^	1.993	1.262/3.150	0.003

^†^ Presence of diabetes, ^‡^ greater than or equal to 10 medicines consumption and ^§^ female. The variables including in the model were: cohabitation, social assessment, mental assessment, Barthel Index, GADS, main pathologies, emergency room, number of medicines, number of prescribers. Forward methods was used for the analysis.

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
