# Peer review of "Evaluation of Inappropriate Prescribing in Patients Older than 65 Years in Primary Health Care"

_jcm, 2019, doi:10.3390/jcm8030305_

Reviewer 1 Report

The study adds to the growing literature on potentially inappropriate medications prescribed to the elderly.  The manuscript is well-written, interesting and contain information relevant to the prescribing of medicines for the elderly. The methods, although descriptive, is parsimonious and transparent.

Minor Comments

A more complete Table 3 would have been useful to allow the reader to ascertain the magnitude and sign of all the independent variables

It would be interesting and informative (in the future) to estimate the model in Table 3, this time with a count dependent variable to see whether the results/estimates are stable

It seems temporal variation (not clear the manuscript text) captured by a year dummy, interactions and potential endogeneity of some of the right-hand side variables should have been accounted for in the logit model. Ignore the latter, if the methods used are designed to convey descriptive information.

As with all studies, there should be some limitations to the study due to, for example, methods chosen, data used, etc. The authors failed to acknowledge any limitations in the study.

Should “Of note was that patients in our sample had a 60% risk of anxiety and a 32.2% risk of depression.” Be “Of note was that 60% and 32.2% of patients in our sample had risk of anxiety and depression, respectively.”?

Should these GADS (categories) % add up to 100%?

I could not trace 45.2% in the text “Inappropriate prescribing was detected in 45.2% of the participants…” to Table 1. A good practice would be to write in the text values/numbers that are also in the tables, otherwise the authors can state in parenthesis (not reported)

Please check this statement “With regard to pathologies, inappropriate prescribing was most frequent among patients with a diagnosis of diabetes (Table 2).” to Table 2.

Author Response

In the attached document are the responses to the comments of the reviewer.

Linguistic revision of the manuscript has been requested

Reviewer 2 Report

General comments

In this manuscript, the authors aim to determine the frequency of inappropriate prescribing by primary care physicians and its predisposing factors in patients who used multiple prescription drugs (5 or more) and were older than 65 years. In a cross-sectional study of 425 patients recruited from primary care centers in Spain, the authors describe potentially inappropriate prescribing (PIP) as defined by STOPP/START criteria as related to the possible predisposing factors sociodemographic characteristics, clinical characteristics and medication use. The main finding was that the majority of the population of this patient group had at least one PIP, and having PIP was directly related to the number of prescribed medications, gender, and specific pathologies of the individual.

The strength of this study is that it addresses a clear important clinical question with a very thorough data collection.

The major weaknesses of this manuscript are the reporting and discussion of the methodology.

Specific comments

Introduction

Very nice argument for the gap addressed by this study. The aim is clear.

Methods

Line 82-85: Introduction of bias by excluding psychiatrically ill or patients with dementia. Theoretically these groups are more likely to have PIP.

Line 98: Please define the STOPP/START criteria in the methods section. These criteria are complicated. How were the criteria applied – i.e. was one investigator responsible for coding? With these complex variables it would  check for inter-researcher validity in application of the criteria?

Line 99-106: Please give more specific information on how the following variables were defined:

·       diseases and disorders

·       emergency room visits

·       Number of medicines

·        number of prescribing doctors

Line 114: “Multivariate analysis was done with binary logistic regression.” It is unclear how you chose the predictors for the model. How was the model built?

Line 117: I would expect a document identifier for the ethics committee decision.

Results

Table 1: The GADS is not diagnostic – it can be used to predict risk for potential clinically relevant symptoms. The terminology should be changed in the table. Similarly, the terms describing the Gijon scale are confusing.

Line 158: In the results, you show that female gender, diagnosis of diabetes and use of more than 10 medicines were directly related to inappropriate prescribing (Table 3). However, in the abstract you claim both diabetes and depression are related to having PIP. This is confusing.

Table 3 is confusing. If you have done a multivariate analysis I would like to see the model. Also, I assume you report the odds ratio?

Discussion

I would like to see a discussion of your methodology including:

·       Discussion of the potential bias generated by your inclusion and exclusion criteria.

·       Discussion of the validity of the STOPP/START criteria

·       Discussion of your choice of variable assessment tools:  GADS, social risk assessment by (Gijón Scale), adherence by MGLS, functional assessment by Barthel Index, cognitive assessment by Pfeiffer test, quality of life by the ED-5D. Also, why not use the EQ-5D domain to assess depressive symptoms (it is validated for this!) instead of less studied GADS?

·       Discussion of the external validity of odds ratio as a measurement

Line 166-167: “We used these criteria together with review of our participants’ medical records and the contents of their medicine cabinets, an approach which avoided overestimation of PIP.” This is very interesting but the process should be more clearly explained in the methods section.

Conclusions

Line 208-209: “In addition, PIP was directly related to the number of prescribed medicines, gender, and certain pathologies (e.g. diabetes and depression).” I am still uncertain how depression was defined – if you only use GADS you cannot claim to have a diagnosis, and of course in the logistic regression you only report a relationship with diabetes – please clarify.

Author Response

(The authors gave the same response as above.)
